# E7 Peptide Enables BMSC Adhesion and Promotes Chondrogenic Differentiation of BMSCs Via the LncRNA H19/miR675 Axis

**DOI:** 10.3390/bioengineering10070781

**Published:** 2023-06-30

**Authors:** Weili Shi, Jiangyi Wu, Yanbin Pi, Xingran Yan, Xiaoqing Hu, Jin Cheng, Huilei Yu, Zhenxing Shao

**Affiliations:** 1Peking University Third Hospital, Beijing Key Laboratory of Sports Injuries, Department of Sports Medicine, Institute of Sports Medicine of Peking University, Beijing 100191, China; shiweilixmu@bjmu.edu.cn (W.S.); piyb@vip.sina.com (Y.P.); huxiaoqingbd01@sina.com (X.H.); rubyeyre@163.com (J.C.); 2Plastic Surgery Hospital, Chinese Academy of Medical Sciences and Peking Union Medical College, Beijing 100144, China; wjy4199@163.com; 3State Key Laboratory of Biochemical Engineering, Institute of Process Engineering, Chinese Academy of Sciences, Beijing 100190, China; yanxingran15@mails.ucas.ac.cn; 4University of Chinese Academy of Sciences, Beijing 100049, China

**Keywords:** mesenchymal stem cells, E7 peptide, chondrogenic differentiation, H19/miR-675, Konjac Glucomannan microsphere

## Abstract

Therapeutic strategies based on utilizing endogenous BMSCs have been developed for the regeneration of bone, cartilage, and ligaments. We previously found that E7 peptide (EPLQLKM) could enhance BMSC homing in bio-scaffolds and, therefore, promote cartilage regeneration. However, the profile and mechanisms of E7 peptide in cartilage regeneration remain elusive. In this study, we examined the effect of E7 peptide on the BMSC phenotype, including adhesion, viability and chondrogenic differentiation, and its underlying mechanism. The konjac glucomannan microsphere (KGM), a carrier material that is free of BMSC adhesion ability, was used as the solid base of E7 peptide to better explore the independent role of E7 peptide in BMSC behavior. The results showed that E7 peptide could support BMSC adhesion and viability in a comparable manner to RGD and promote superior chondrogenic differentiation to RGD. We examined differentially expressed genes of BMSCs induced by E7 compared to RGD. Subsequently, a real-time PCR validated the significantly upregulated expression of lncRNA H19, and the knockdown of lncRNA H19 or miR675, a downstream functional unit of H19, could significantly obscure the chondrogenic differentiation induced by E7. In conclusion, this study confirmed the independent role of E7 in the adhesion and viability of BMSCs and revealed the pro-chondrogenic effect of E7 on BMSCs via the H19/miR675 axis. These results could help establish new therapeutic strategies based on employing endogenous BMSCs for cartilage tissue regeneration.

## 1. Introduction

The therapeutic strategy utilizing mesenchymal stem cells (MSCs) for tissue regeneration has been explored widely and shown huge progress [1]. Endogenous MSC homing is one of the most promising strategies among such research [2]. This strategy can take advantage of the positive effects of MSCs in tissue regeneration, while avoiding the shortcomings of exogenous MSCs, such as the requirement of rigorous standard of in vitro cell management, immunoreaction, heterogeneity, and potential pro-neoplasmic effect [3]. However, the utilization of endogenous MSCs still needs to be promoted because MSCs are usually located at a given niche and stay in a silent status before being appropriately activated [2]. Additionally, the distribution of functional MSCs is not extensive in all kinds of tissues; for example, cartilage tissues lack MSCs with adequate regeneration ability [4]. The chondrogenesis of MSCs featured in the expression of genes, including col II, acan, and sox9, and the secretion of the extracellular matrix could contribute to cartilage regeneration [5]. Thus, the efficient strategy of homing MSCs to an injury region of cartilage and promoting MSC chondrogenic differentiation has been the research direction in a number of studies investigating cartilage regeneration [6].

We previously discovered a BMSC (bone-marrow-derived mesenchymal stem cell)- affinity peptide E7 (Glu-Pro-Leu-Gln-Leu-Lys-Met, EPLQLKM) and applied it in the regeneration of cartilage [7]. Firstly, microfracture was set to help release BMSCs to the cartilage defect area, and then E7-bonded composite material was placed into the cartilage defect to promote the homing of BMSCs in the area. When composited with several kinds of materials, including polycaprolactone electrospun mesh [7], chitosan hydrogel [8], demineralized bone matrix [9], and silk-fibroin–gelatin scaffold [10], E7 peptide could significantly increase the amount of BMSCs gathered in the cartilage defect area and improve the cartilage regeneration effect. However, whether E7 peptide could serve as an adhesion target for BMSCs as an individual factor in a composite material without the pre-existing cell-adhesion ability of the base material and whether E7 peptide could have an impact on the phenotype of BMSCs after being adhered are still unclear. 

In the present study, we employed konjac glucomannan microsphere (KGM), a basic material without BMSC adhesion ability, as the scaffold to provide the solid base for E7 peptide. KGM has previously been used as a cell microcarrier with electric charge modification [11]. In this study, we linked E7 peptide to the surface of KGM (E7-KGM) to elucidate the role of E7 peptide in BMSC adhesion and phenotype regulation since the use of KGM could exclude the effect of issues external to E7 peptide, such as the scaffold itself, on BMSC adhesion. RGD peptide (Arg-Gly-Asp), a well-known peptide that adheres to cells via integrin [12], was set as the positive control of BMSC adhesion (RGD-KGM). Through the measurements of the amount, viability, and phenotype of BMSCs that adhered to free KGM, E7-KGM, and RGD-KGM, we found the positive role of E7 peptide in the function of BMSC adhesion and chondrogenic differentiation of BMSCs when compared to RGD peptide. These findings could be utilized to enhance the effectiveness of endogenous BMSC homing and promote the chondrogenic differentiation effect of BMSCs in cartilage tissue engineering. This may offer valuable clues for developing novel strategies for cartilage regeneration and repair. 

## 2. Results 

### 2.1. E7 Peptide Could Be the Adhesion Site for BMSCs

KGMs with an diameter of 150 - 200 µm were prepared according to a previous study. When E7 peptide was added into the system, the sulfhydryl group of E7 peptide combined with the epoxy group on the surface of KGMs, leading to the formation of E7-coated KGMs (E7-KGM) (Figure 1A). When co-cultured with BMSCs, the surface of E7-KGMs was full of adhered BMSCs, and the microspheres were linked via cell colonies, similar to that of electric-charge-modified KGMs (EC-KGM) and RGD-coated KGMs (RGD-KGM), while free KGMs (Blank) showed rare adhered BMSCs. The percentages of cell adhesion were 29.58 ± 7.54% for E7-KGM, 36.26 ± 13.17% for RGD-KGM, and 7.92 ± 2.53 for Blank KGM. Because free KGMs showed little cell adhesion ability, as expected, we found that E7 peptide could serve as an individual cell adhesion target and be composed with materials for bioactive cell adhesion regardless of the cell adhesion ability of the materials. To test the role of special amino acid sequences in the biological function of E7 peptide, we used M7 peptide (MLKPLEQ, a scrambled peptide with the same amino acid constituents but a different amino acid sequence from peptide E7) to modify KGMs and found few adhered BMSCs (10.00 ± 3.31%), similar to the result for free KGMs (Figure 1B). Thus, we demonstrated that E7 peptide with a unique amino acid sequence could serve as the adhesion target of BMSCs independent of any other adhesion target. 

### 2.2. E7 Peptide Helps Maintain the Viability of BMSCs

The effects of E7 peptide on cell behaviors, including viability, proliferation, and differentiation, are also important issues to be elucidated before the biological application of E7 peptide in tissue regeneration. To explore the viability of BMSCs that adhered to KGMs exclusively via E7 peptide, we tracked the cell status across a culture period of two weeks. The surface of E7-KGMs was full of cells from day 3 to day 14; colonies of KGMs formed through the linkage constituted by the cells at day 3, magnified gradually up to day 8, and then were maintained to day 14. At day 6, using a scanning electron microscope (SEM), we found that the cells on the surface of E7-KGMs spread well and showed abundant microvilli, indicating good cell viability, whereas the cells on the surface of EC-KGMs and M7-KGMs were obviously less and had poor microvilli (Figure 2A). The live/dead staining test further confirmed the viability of the cells on the surface of E7-KGMs (Figure 2B). The Alamar blue test showed that, at the timepoints of 24 h, 48 h, and 72 h after the seeding of cells, the viability of the E7-KGM and RGD-KGM group was significantly higher than that of the EC-KGM groups (Figure 2C). Thus, E7 peptide could facilitate the viability of BMSCs similar to RGD peptide.

### 2.3. E7 Peptide Promotes the Chondrogenic Differentiation of BMSCs

E7 peptide has been used in several composite materials for cartilage regeneration, although the role of E7 in the promotion of cartilage regeneration has not yet been elucidated. In this study, we explored whether E7 peptide could affect BMSC chondrogenic differentiation independently. We seeded BMSCs on the surface of E7-KGMs, RGD-KGMs, and EC-KGMs for one, three, and five days. The results showed that the mRNA expressions of sox 9, acan, and col II were significantly higher in the E7-KGM group than in the RGD-KGM and EC-KGM groups (Figure 3A), and the protein expressions of sox 9 and col II were also higher in the E7-KGM group than the RGD-KGM and EC-KGM groups (Figure 3B). When the co-culture system was maintained for 28 days, the BMSCs of E7-KGMs and RGD-KGMs formed cell clumps, while those of the EC-KGM group died. The paraffin section and Alcian blue staining test of these cell clumps showed more glycosaminoglycan formation in E7-KGMs (Figure 3C).

### 2.4. E7 Peptide Affects the mRNA Expression Profile of BMSCs

To further explore the molecular mechanism underlying the effect of E7 peptide on BMSCs, RNA sequencing (RNA-seq) was performed, and the results showed that the gene expression profile of BMSCs on E7 peptide was significantly different from that of EC-KGMs; in particular, 1527 genes were significantly upregulated (*p* < 0.05) in E7-KGMs compared to EC-KGMs, while 1374 genes were significantly downregulated (*p* < 0.05) (Figure 4A). The GO analysis showed that the upregulated and downregulated genes were involved in cell adhesion, inflammatory response, extracellular matrix organization, response to lipopolysaccharide, cell–cell signaling, and cell differentiation. The KEGG pathway analysis of the upregulated genes revealed a significant enrichment of signaling pathways related to focal adhesion, cell cycle, ECM–receptor interaction, metabolism, and chondrogenic differentiation. On the other hand, the downregulated genes were enriched in several inflammatory response pathways, including the NOD-like receptor signaling, NF-kappa B signaling, and Toll-like receptor signaling pathways (Figure 4B). Nevertheless, the RNA expression profile of the E7 group was relatively similar to that of the RGD group, with 37 genes being significantly upregulated (*p* < 0.05) and 35 genes being significantly downregulated (*p* < 0.05) in E7-KGMs compared to RGD-KGMs (Figure 4C). The GO analysis showed that the differentially expressed genes were related to cell–cell signaling, chemokine activity, and CXCR chemokine receptor binding, and the KEGG pathway analysis revealed enrichment of signaling pathways related to cytokine–cytokine receptor interaction and cell adhesion molecules (Figure 4D). Among the genes upregulated in E7-KGMs compared to RGD-KGMs, we found lncRNA H19 to be significantly more highly expressed, and such high expression has been reported to be associated with chondrogenic differentiation; the real-time PCR analysis also validated the higher expression of lncRNA H19 in the E7-KGM group (Figure 5A).

### 2.5. Inhibition of H19/mi-675 Suppresses the Pro-Chondrogenic Effect of E7 Peptide on BMSCs

The co-culture system of BMSCs and E7-KGMs was set for 24 h for the adhesion of BMSCs on E7-KGMs; then, H19 siRNA was added to the co-culture system. At the timepoints of two and four days after the addition of H19 siRNA, the H19 expression of BMSCs was significantly decreased, as expected, while at the same time, the mRNA expressions of chondrogenic differentiation markers, including acan, col II, and sox 9, were all markedly suppressed in the H19 siRNA group (Figure 5B). The protein expressions of acan, col II, and sox 9 were measured and showed a remarkable reduction in the H19 siRNA group (Figure 5C). We further explored whether inhibition of miR-675, one of the downstream factors of H19 confirmed to be able to promote chondrogenic differentiation in chondrocyte [13], could affect the pro-chondrogenic effect of E7-KGMs on BMSCs. When miR-675 was inhibited, the expression of H19 remained stable, while the expressions of acan, col II, and sox 9 were all significantly suppressed (Figure 6A). The protein expressions of acan, col II, and sox 9 showed a remarkable reduction in the miR-675 inhibitor group (Figure 6B).

## 3. Discussion

MSC-based therapy in tissue regeneration has drawn extensive attention in recent research, and the use of MSCs at the local site of a tissue defect has shown huge potential in promoting tissue regeneration [2]. For example, microfracture operation for the treatment of cartilage defect could improve cartilage repair through the release of BMSCs localized in bone marrow to the cartilage defect area [14], and an E7 peptide-combined scaffold seeded in the cartilage defect area could further improve cartilage regeneration via the binding and homing of BMSCs in the cartilage, rather than flowing away [10]. In order to extend the use of E7 peptide in cartilage regeneration, we need more information on the functional profile of E7 peptide in BMSC adhesion and activation. 

The materials used in previous studies for composites with E7 peptide all inherently have a positive cell adhesion ability, including polycaprolactone electrospun mesh, chitosan hydrogel, demineralized bone matrix, and silk-fibroin–gelatin scaffold. Thus, it is unclear whether the role of E7 peptide in the adhesion of BMSCs on a composite material is assistive or independent. An assistant role means that the presence of a cell-adhesion-positive material is a necessary premise for the pro-BMSC-adhesion function of E7 peptide; that is, E7 peptide could augment but not confer cell adhesion ability to materials. An independent role means that E7 peptide could adhere BMSCs without the help of cooperated materials; that is, it could confer pro-BMSC-adhesion ability to materials. Additionally, whether the enhanced cartilage regeneration should be attributed to the accumulation of BMSCs and their interaction with the surrounding microenvironment or whether it is induced, at least partly, by E7 peptide is not clear. In this study, we used KGM as the base material for E7 peptide to exclude the potential effect arising from the cell-adhesion ability of the base material used. KGM-E7 microspheres showed significant BMSC adhesion, while pure KGMs showed no obvious adhesion. Additionally, the negative adhesion of KGM-M7 emphasized the pivotal role of specific E7 peptide sequence in the bioactive cell adhesive function. Thus, E7 peptide could serve as an individual cell-adhesion target and be composed with materials for bioactive cell adhesion regardless of the cell-adhesion ability of the materials used.

A previous research study showed that E7 composite material could promote cartilage regeneration at a cartilage-defect site [7]. The underlying mechanism was speculated to be due to the aggregation of BMSCs at the local site, followed by proliferation and chondrogenic differentiation in the local microenvironment. However, whether E7 peptide played a role in the viability and differentiation of BMSCs that adhered to the base material was not clear. In this study, we found that BMSCs that adhered to KGMs via E7 peptide maintained good viability for at least one month in vitro, similar to RGD. Moreover, the microstructure of BMSCs showed well-spreading and abundant microvilli on the surface of E7-KGMs, and these results were further supported by the Alamar blue test. In terms of chondrogenic differentiation, we found that E7 significantly enhanced the mRNA expressions of chondrogenic differential markers, including col II, acan, and sox 9, which were consistent with the protein expressions of sox 9 and col II. The extracellular matrix of BMSCs cultured on the surface of KGMs showed greater glycosaminoglycan formation, a specific cartilaginous polysaccharide, suggesting the higher chondrogenic induction ability of E7 compared to RGD. 

The underlying mechanism of the difference in cell viability between E7/RGD and EC and chondrogenic differentiation between E7 and RGD could be explored in the data obtained from RNA sequencing. The mRNA expression profile showed a vast difference between E7 and EC but a relatively similar profile between E7 and RGD. Considering the mechanism underlying the electrostatic adherence in EC and integrin–peptide interaction in RGD [15], the significant mRNA expression discrepancy may be due to completely different adherence mechanisms, that is, due to the class of electrostatic adherence and the class of peptide–domain interaction. The relatively minor difference in the mRNA expression profile between E7 and RGD suggests that the mechanism of E7 peptide adherence may belong to or occur adjacent to peptide–domain interaction, but it is definitely different from that of the RGD peptide [16].

LncRNA H19 is well-known in tumorigenesis as a tumor suppressor [17], and recently, the role of H19 in other pathologies has been noticed, including atherosclerosis [18], osteoporosis [19], fibrosis [20], and ischemic stroke [21]. lncRNA H19 acts as a metabolic correlate in cartilage and cultured chondrocytes [22], and the overexpression of H19 could promote the chondrogenic differentiation of hUCMSCs [23]. Along with the clue from the screening of H19 in E7-KGMs via RNA-seq, we validated the significantly higher expression of H19 in the E7-KGM group compared to the RGD-KGM and EC-KGM groups. Furthermore, the chondrogenic differentiation-promoting effect of E7-KGMs on BMSCs was negated when H19 expression was knocked down by H19 siRNA. It has been reported that H19 is a primary miRNA precursor for microRNA-675, which is transcribed from the first exon of H19 [24] and serves as the functional unit of H19 in several biological processes, such as tumorigenesis, invasion [25], and cardiomyopathy [26]. In this study, we proved that inhibition of miR-675 significantly suppressed the elevated chondrogenic differentiation of BMSCs induced by E7-KGMs. The findings in the present study suggest that the H-19/miR-675 axis participates in the BMSC chondrogenic differentiation motivated by E7 peptide.

The usage of KGMs in biological applications has mainly been as a means of bioactive assistance through charge modification; however, the shortcomings of such usage are becoming increasingly apparent, including the limitation of bio-interaction with the adhered cells and the absence of persistent cell phonotype support [27]. Here, we demonstrated that peptide modification could confer favorable bioactive functions to KGMs, including cell adherence, cell viability, cell proliferation, and even differentiation phenotype. The KGMs used in this study were solid, and BMSCs could grow on their surface; nevertheless, these KGMs supported a 3D environment for BMSCs because cells were distributed on multiple planes, including on the horizontal, vertical, and various transverse planes, and these cells were interlaced in the interspace of KGMs. Thus, the cell behavior observed in this study may be more closer to in vivo cell status compared to that cultured in a flat dish [28].

There are several limitations to this study. Firstly, the primary target regarding how BMSCs directly interact with E7 peptide is not yet clearly elucidated, thus obscuring the understanding of the mechanism of E7 peptide function. Secondly, we showed the mRNA expression profile and several signaling pathways of BMSCs that are induced by E7 peptide, and we revealed that lncRNA-H19/miR-675 mediates the pro-chondrogenic effect of E7 peptide, but the upstream and downstream signals in the chondrogenic differentiation process of BMSCs remain to be further elucidated. 

## 4. Materials and Methods

### 4.1. Isolation and Identification of BMSCs

Rat BMSCs were isolated according to previous reports [29]. Briefly, bone marrow was obtained from Sprague Dawley rats weighing 80 g by isolating it from the femur and tibia. The cells were then incubated in α-minimal essential medium (α-MEM) supplemented with 10% fetal bovine serum (FBS), 100 U/mL of penicillin, and 100 mg/mL of streptomycin. The incubation was carried out at 37 °C with 5% humidified CO_2_. After 3 days of incubation, the non-adherent cells were removed by changing the culture medium. Upon reaching confluence after 4 to 5 days of culture, the adherent cells were defined as passage 0.

The specific cell-surface antigen markers of BMSCs were examined using flow cytometry (FCM). For this analysis, passage 2 cells were utilized, and the primary antibodies included anti-CD44 (eBioscience, San Diego, CA, USA, 12-0444-82), anti-CD90 (BD, NJ, USA, 561973), CD34 (Abcam, Cambridge, UK, ab81289), and CD45 (BD, 561867). To determine the adipogenic, osteogenic, and chondrogenic differentiation potentials of BMSCs, a trilineage-induced differentiation assay was conducted. The experiments involved passage 2. In brief, the cells were seeded at a density of 1 × 10^5^ cells/well in a six-well plate. Adipogenesis and osteogenesis induction were achieved using Rat MSC Adipogenic and Osteogenic Differentiation Media (Cyagen Biosciences, Guangzhou, China), respectively. Following a 3-week culture period, oil red O staining was performed to examine adipogenesis, while alizarin red staining was used for osteogenesis assessment. For chondrogenesis, a pellet culture technique was employed. Briefly, the cells were digested with trypsin, and a total of 1 × 10^6^ cells/tube was washed with α-MEM twice, resuspended in 0.5 mL of Rat MSC Chondrogenic Differentiation Medium (Cyagen Biosciences) in a 15 mL polypropylene centrifuge tube, and then centrifuged at 150 g for 5 min. The pellet was incubated at the bottom of the tube with the supernatant at 37 °C in 5% CO_2_ for 24 h, and then the tube was gently flicked to ensure that the pellet was free floating. The medium was changed every 2–3 days. After three weeks of incubation, the pellet was fixed in 4% (*m*/*v*) paraformaldehyde, embedded in paraffin, and subjected to Alcian blue staining to assess glycosaminoglycan formation in the extracellular matrix (ECM) of the pellet.

### 4.2. Preparation of KGM Microspheres

KGM microspheres were prepared via water-in-oil (W/O) emulsion, as described previously [30]. The formed microspheres contained 12% KGM and 25% crosslinker. They were washed with petroleum ether, ethanol, and deionized water successively. Microspheres with diameters between 150 and 200 μm were sieved out and used for further modification. Electric-charge-modified KGM microcarriers were prepared by an amination process as follows: 10 g of vacuum-dried KGM microspheres was mixed with 6 g of 35% (*w*/*w*) NaOH solution and stirred at 70 °C with 250 rpm for 1 h. Then, DEAE·HCl was added at the ratio of 3:5 (*w*/*w*) to microsphere weight to form KGM microcarriers with amino group on the surface. The reaction mixture was stirred for 4 h at 70 °C, and the formed KGM microcarriers were washed with deionized water, incubated with HCl solution (pH 2) overnight, and finally washed with deionized water until the pH value of the eluate achieved a neutral value. To link peptides to the surface of microspheres, the epoxide group was first linked to activate KGMs. A total of 6 g of vacuum-dried KGM microspheres was mixed with 5 mL of Oxirane,2,2’-[1,4-butanediylbis(oxymethylene)]bis-, 10 mL of 0.5 M NaOH, and 10 mg of NaBH_4_; stirred at 50 °C with 120 rpm for 4 h; and then washed with ethanol 3 times and with deionized water 3 times. The activated KGMs were achieved and then preserved in 20% ethanol. A total of 0.2 g of vacuum-dried activated KGMs was mixed with 0.6 mL of Na_2_CO_3_/NaHCO_3_ (pH = 10) and 10 mg of peptides (E7, M7 (Scrambled peptide, MLKPLEQ), RGD) and stirred at 50 °C with 120 rpm for 5 h; afterward, the supernatant was discarded, ice water was added for 1–2 min, and the pellet was washed with PBS 3 times. Then, 0.6 mL of PBS and 0.05 mL of glutaraldehyde were added; the mixture was stirred at 50 °C with 120 rpm for 1 h and washed with deionized water for 3 times and PBS for 3 times; and, finally, peptide-KGMs were obtained and preserved in PBS.

### 4.3. Co-Culture of BMSCs and KGMs

To evaluate the adhesion of BMSCs on KGMs, 50 µL of KGMs and 1 × 10^5^ BMSCs were seeded in a free-adhesion 6-well plate, α-MEM medium was replaced every 2 days, and photos were taken under an optical microscope. For calculating the percentages of cell adhesion, we shook the culture dish gently to make the unadhered BMSCs equably suspended in the medium after co-culture of BMSCs and KGMs in serum-free culture medium for 12 h. Then, we transferred all of the medium with unadhered BMSCs suspending inside to perform the cell counting in the cell counting plate, and the number of unadhered BMSCs in the medium was calculated and counted as N. The cell adhesion rate on the scaffolds was calculated by the following formula: (total cell number–N)/total cell number 100%.

Small interfering RNAs (siRNAs) against lncRNA H19 (named si H19) were designed and synthesized via RiboBio. The sequence of the functional si-H19 was GGATGACAGGTGTGGTCAA. For miRNA transfection, miR-675 inhibitors and negative control (RiboBio) were transfected into the cell samples, using Lipofectamine 3000 (Invitrogen) according to the manufacturer’s instructions. The cell samples were collected at designated times for the following tests.

### 4.4. SEM

The KGM samples with BMSCs on the surface were fixed with 2.5% (*v*/*v*) glutaraldehyde buffered with PBS; dehydrated using a graded series of ethanol washes; and dried to a critical point (EM CPD300; Leica, Wetzlar, Germany), using carbon dioxide (CO_2_). The samples were sputter-coated with gold prior to SEM observation. BMSCs on the surface of KGMs were observed using SEM (S-4800 field-emission scanning electron microscope; Hitachi, Tokyo, Japan).

### 4.5. Live/Dead Staining

To assess cell viability, BMSCs on E7-KGMs were subjected to LIVE/DEAD staining (Invitrogen, Carlsbad, CA, USA), using confocal microscopy. After 5 days of co-culture, the samples were immersed in a 1 mL working solution containing 2 mM of calcein-AM and 4 mM of ethidium homodimer-1 reagents. Incubation at room temperature for 1 h was allowed for staining. A confocal microscope with excitation wavelengths of 488 nm or 568 nm was utilized to visualize calcein AM (green fluorescence: labeling live cells) and ethidium homodimer-1 (red fluorescence: labeling dead cells). 

### 4.6. Alamar Blue Assay

To evaluate the cell viability on differentially modified KGMs, 2 µL of KGMs and 5 × 10^3^ BMSCs were seeded in a free-adhesion 96-well plate. After 24, 48, and 72 h of co-culture, the medium was replaced with low-serum α-MEM containing 10% Alamar blue reagent and incubated for 2 h; then, the medium was transferred to a 96-well plate. Fluorescence at an excitation wavelength of 540 nm and an emission wavelength of 590 nm was measured. The background signal was determined using the negative control of the medium alone, without cells. The % reduction of Alamar blue reagent was calculated using the fluorescence readings, according to the manufacturer’s instructions.

### 4.7. Real-Time Quantitative PCR

Total RNA was isolated from BMSCs, using TRIzol (Invitrogen, CA, USA). Subsequently, 1 µg of RNA was reverse transcribed using the Thermo Scientific RevertAid First Strand cDNA Synthesis Kit (Thermo Fisher, MA, USA). The resulting cDNA was then amplified using a real-time PCR system (Applied Biosystems) with SYBR^®^ Select Master Mix (Applied Biosystems, CA, USA). The primers used for amplification of type 2 collagen (col II), SRY-box transcription factor 9 (sox 9), and aggrecan are provided in Table 1. The PCR cycling condition involved an initial denaturation at 95 °C for 30 s, followed by 40 cycles of amplification consisting of 15 s of denaturation at 95 °C and 30 s of extension at 60 °C. To determine the relative mRNA expression levels of the target genes, normalization against glyceraldehyde 3-phosphate dehydrogenase (GAPDH) was performed. The calculation was carried out using the comparative CT method.

### 4.8. Western Blotting

The BMSC samples were lysed using RIPA lysis buffer, followed by SDS–polyacrylamide gel electrophoresis (PAGE) for resolution and transfer to PVDF membranes. Afterward, the membranes were incubated overnight at 4 °C with the respective primary antibodies. Horseradish-peroxidase-conjugated secondary antibodies were applied for 1 h at room temperature. The membranes were visualized using the BIO-RAD ChemiDoc XRS+ system. Anti-Col II (Abcam, ab188570, 1:2000), anti-sox 9 (Abcam, ab185866, 1:1000), anti-acan (Thermo Fisher, PA1-1746, 1:1000), anti-β-actin (ZSGB-BIO, TA-09, 1:1000), anti-mouse secondary antibody (ZSGB-BIO, ZB-2305, 1:4000), and anti-rabbit secondary antibody (ZSGB-BIO, ZB-2301, 1:4000) were used.

### 4.9. Histology 

The specimens for H&E staining and Alcian blue staining were immediately fixed in 10% neutral buffered formalin, dehydrated through an alcohol gradient, and then cleaned and embedded in paraffin blocks. Histological sections (5 µm) were prepared using a microtome and stained with hematoxylin and eosin (H&E). Alcian blue staining was performed according to standard procedures to examine the general appearance of proteoglycan. All sections were analyzed by 2 pathologists, who were blinded to the treatment groups.

### 4.10. RNA Sequencing

After 3 days of co-culture, BMSCs from different scaffolds were collected for RNA sequencing analysis, using the NovelBrain Cloud Analysis Platform. Total RNA was extracted from BMSCs, using TRIzol reagent (Invitrogen). Construction of cDNA libraries was carried out for each pooled RNA sample, using the VAHTSTM Total RNA-seq (H/M/R) kit. Differential gene and transcript expression analysis were performed using TopHat and Cufflinks. Gene and lncRNA counts were obtained using HTseq. Gene expression levels were determined using the FPKM method. We applied the DESeq algorithm to calculate differentially expressed genes. Significant analysis was performed using the *p*-value and a false discovery rate (FDR) analysis. Differentially expressed genes were identified with fold change > 2 or fold change < 0.5, and FDR < 0.05. Furthermore, a GO analysis was performed to facilitate the elucidation of the biological implications of differentially expressed genes, including biological process (BP), cellular component (CC), and molecular function (MF) [31]. The GO annotations from NCBI (http://www.ncbi.nlm.nih.gov/, accessed on 11 May 2018), UniProt (http://www.uniprot.org/, accessed on 11 May 2018), and the Gene Ontology (http://www.geneontology.org/, accessed on 11 May 2018) were downloaded. Fisher’s exact test was applied to identify significantly enriched GO categories. Pathway analysis was used to identify significantly enriched pathways affected by differentially expressed genes according to the Kyoto Encyclopedia of Genes and Genomes (KEGG) database [32]. Fisher’s exact test was used to select significantly enriched pathways. Additionally, the threshold of significance was defined by the *p*-value [33]. A pathway activity network was constructed using Cytoscape for graphical representations of significantly enriched biological pathways (*p* < 0.05), including upregulated and downregulated pathways. The data are included in the Appendix A. 

### 4.11. Statistical Analysis

All quantitative data were presented as mean ± SD. Student’s *t*-test was used for comparison between two groups, and one-way ANOVA was used for comparison among three or more groups. A *p*-value <0.05 was considered to be statistically significant. 

## 5. Conclusions

In summary, in this paper, we reported the independent role of E7 in BMSC adhesion and viability, which was found to be comparable to that of RGD peptide, by employing KGMs. Furthermore, we discovered that E7 peptide could exert pro-chondrogenic effects on BMSCs via the H19/miR675 axis, suggesting the superior application potential of E7 peptide in cartilage regeneration compared to RGD peptide. The findings in this work prove that E7 peptide could be composited with any incorporable materials to fabricate chondrogenic biomaterials without considering the inherent cell adhesion ability of the base materials used; thus, the findings could promote cartilage regeneration research employing endogenous BMSCs.

## Figures and Tables

**Figure 1 bioengineering-10-00781-f001:**
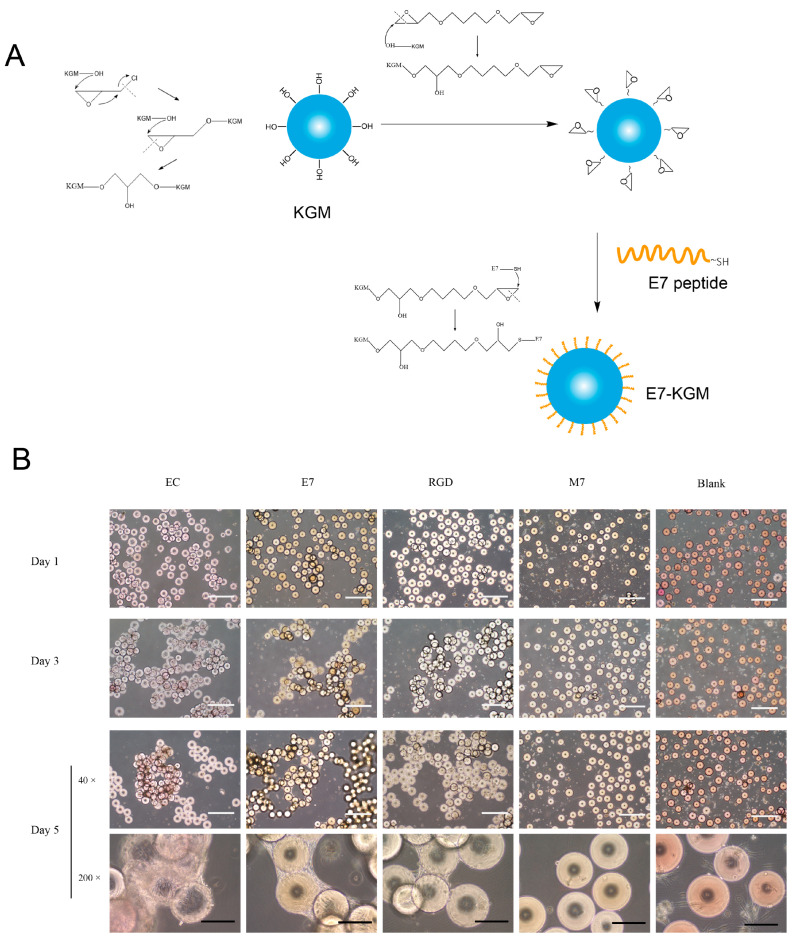
(**A**) Scheme illustrating the preparation process of E7-KGMs. (**B**) BMSCs were co-cultured with EC-KGMs, E7-KGMs, RGD-KGMs, M7-KGMs, and free KGMs for 1, 3, and 5 days, and the adhesion of BMSCs on KGMs was observed under an optical inverted microscope (*n* = 4). White scale bar = 1 mm, and black scale bar = 200 µm. EC-KGMs, electric-charge-modified konjac glucomannan microspheres; E7-KGMs, E7 (EPLQLKM)-modified konjac glucomannan microspheres; RGD-KGMs, RGD peptide (Arg-Gly-Asp)-modified konjac glucomannan microspheres; M7-KGMs, M7(MLKPLEQ)-modified konjac glucomannan microspheres; Blank, free KGMs.

**Figure 2 bioengineering-10-00781-f002:**
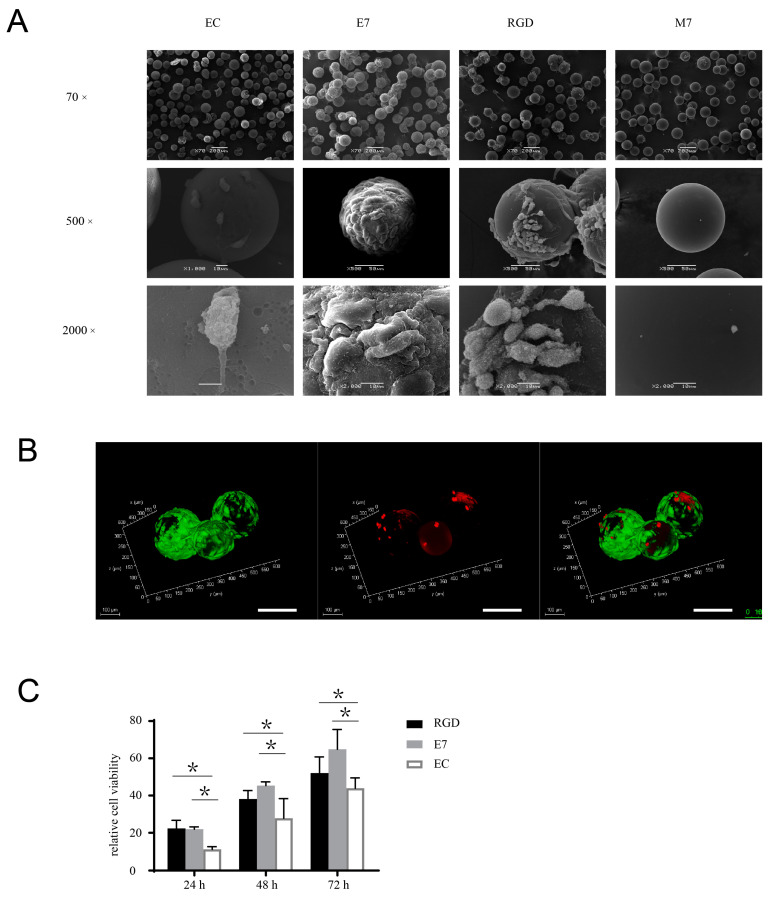
E7 peptide supports the viability and proliferation of BMSCs on the surface of KGMs. (**A**) BMSCs were co-cultured with EC-KGMs, E7-KGMs, RGD-KGMs, and M7-KGMs for 5 days, and the micromorphology of BMSCs on KGMs was observed using SEM (*n* = 3). (**B**) Viability of BMSCs co-cultured with E7-KGMs and RGD-KGMs for 5 days, as assessed using live/dead fluorescence staining under confocal microscope. Green represents live BMSCs, and red represents dead BMSCs. Scale bar = 10 µm. (**C**) Cell viability assessed using Alamar blue staining (*n* = 5, * *p* < 0.05).

**Figure 3 bioengineering-10-00781-f003:**
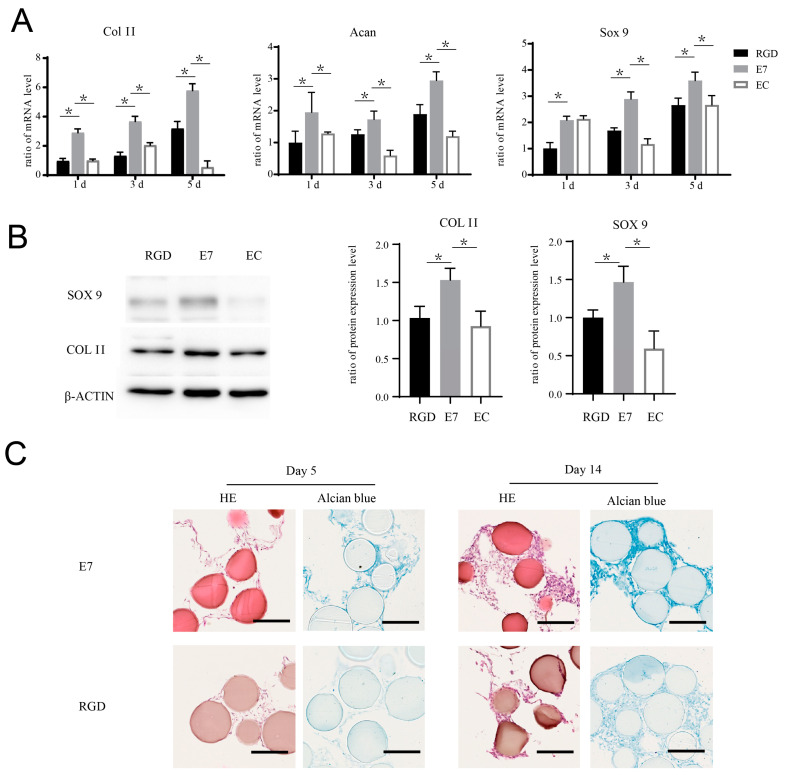
E7 peptide promotes the chondrogenic differentiation of BMSCs on the surface of KGMs. (**A**) BMSCs were co-cultured with EC-KGMs, E7-KGMs, and RGD-KGMs for 1, 3, and 5 days, and the mRNA expression levels of col II, acan, and sox 9 were assessed using real-time PCR (*n* = 3). (**B**) BMSCs were co-cultured with EC-KGMs, E7-KGMs, and RGD-KGMs for 5 days, and the protein expression levels of col II and sox 9 were assessed using Western blotting (*n* = 3). (**C**) BMSCs were co-cultured with E7-KGMs and RGD-KGMs for 5 or 14 days, and the extracellular matrix was measured using H&E staining and Alcian blue staining under optical inverted microscope. Scale bar = 200 μm. * *p* < 0.05.

**Figure 4 bioengineering-10-00781-f004:**
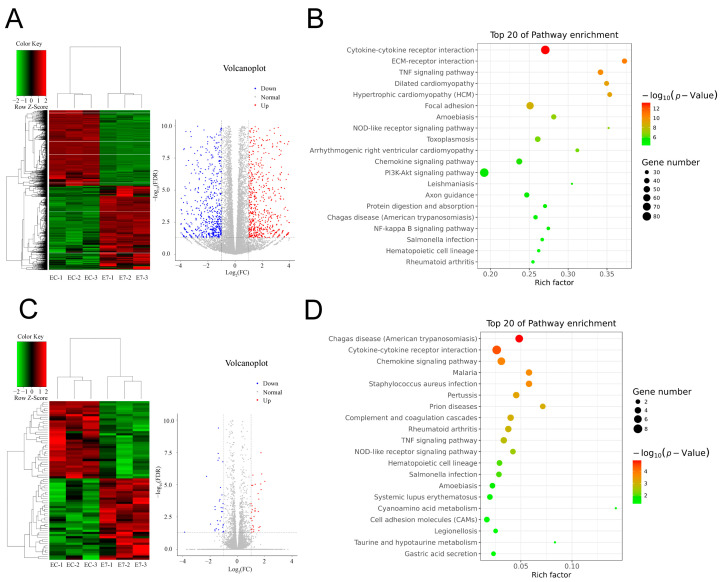
E7 peptide affects the transcriptome of BMSCs. BMSCs were co-cultured with EC-KGMs, E7-KGMs, and RGD-KGMs for 3 days, and the transcriptome was analyzed using RNA sequencing. (**A**) Differentially expressed genes between E7-KGMs and EC-KGMs. (**B**) Pathway analysis of differentially expressed genes between the E7-KGM and EC-KGM groups (*n* = 3). (**C**) Differentially expressed genes between E7-KGMs and RGD-KGMs. (**D**) Pathway analysis of differentially expressed genes between the E7-KGM and RGD-KGM groups (*n* = 3).

**Figure 5 bioengineering-10-00781-f005:**
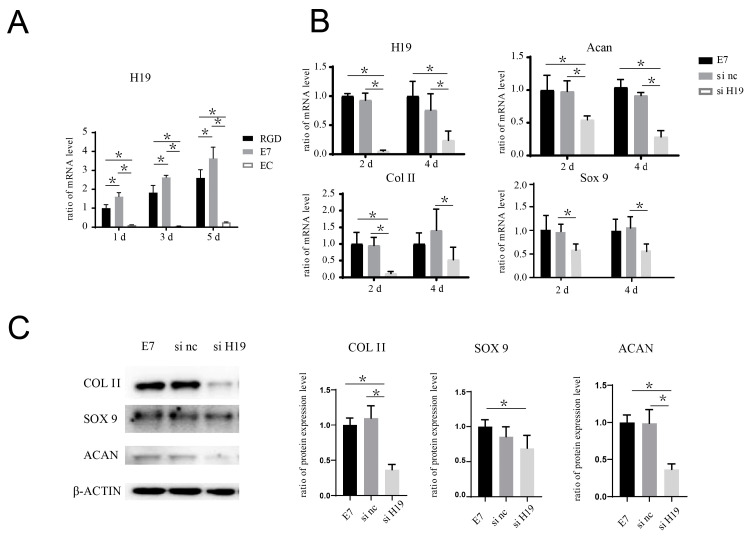
H19 siRNA inhibits the pro-chondrogenic effect seen in BMSCs seeded on the surface of E7-KGMs. (**A**) H19 expression of BMSCs co-cultured with EC-KGMs, E7-KGMs, and RGD-KGMs for 1, 3, and 5 days, as assessed using real-time PCR (*n* = 3). (**B**) H19 siRNA was added in the co-culture system of E7-KGMs and BMSCs, and the mRNA expressions of H19, acan, col II, and sox 9 at days 2 and 4 were assessed using real-time PCR (*n* = 3, * represents *p* < 0.05). (**C**) H19 siRNA was added to the co-culture system of E7-KGMs and BMSCs, and the protein expression levels of acan, col II, and sox 9 at day 4 were assessed using Western blotting (*n* = 3). E7 represents the E7-KGM group, si nc represents the negative/control siRNA group, and si H19 represents the H19 siRNA group. * *p* < 0.05.

**Figure 6 bioengineering-10-00781-f006:**
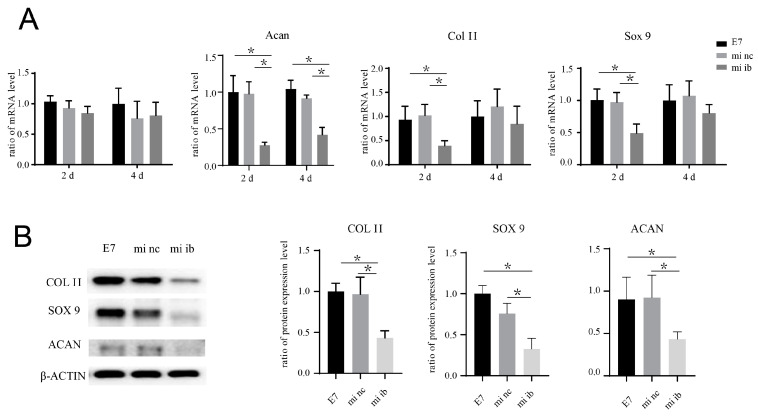
miRNA-675 inhibitor suppresses the pro-chondrogenic effect of E7-KGMs on BMSCs. (**A**) miRNA-675 inhibitor was added to the co-culture system of E7-KGMs and BMSCs, and the mRNA expressions of H19, acan, col II, and sox 9 were assessed using real-time PCR (*n* = 3). * *p* < 0.05. (**B**) miRNA-675 inhibitor was added to the co-culture system of E7-KGMs and BMSCs, and the protein expressions of acan, col II, and sox 9 were assessed using Western blotting (*n* = 3). E7 represents the E7-KGM group, mi nc represents the negative/control miRNA group, and mi ib represents the miRNA-675-inhibitor group. * *p* < 0.05.

**Table 1 bioengineering-10-00781-t001:** Primer sequences used for quantitative RT-PCR.

Gene	Primer Sequence
GAPDH	F: GCAAGTTCAACGGCACAG
	R: GCCAGTAGACTCCACGACA
ACAN	F: CCACTGGAGAGGACTGCGTAG
	R: GGTCTGTGCAAGTGATTCGAG
Sox-9	F: AGGAAGCTGGCAGACCAGTA
	R: ACGAAGGGTCTCTTCTCGCT
Col II	F: CACCGCTAACGTCCAGATGAC
	R: GGAAGGCGTGAGGTCTTCTGT

## Data Availability

Data available on request due to restrictions, eg privacy or ethical The data presented in this study are available on request from the corresponding author.

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
