# Peer review of "E7 Peptide Enables BMSC Adhesion and Promotes Chondrogenic Differentiation of BMSCs Via the LncRNA H19/miR675 Axis"

_bioengineering, 2023, doi:10.3390/bioengineering10070781_

Round 1
Reviewer 1 Report
Peer-review bioengineering – 2416243
The research manuscript entitled “Peptide E7 enables the BMSCs Adhesion and Promote the Chondrogenic Differentiation of BMSCs Via LncRNA H19/miR675 axis” seems an interesting study on the role of peptide E7 in the adhesion, proliferation and chondrogenic differentiation. The manuscript topic fits well within the scope of the journal Bioengineering (ISSN 2306-5354). However, in this current state, the manuscript cannot be considered for publication.
Issues:
1. The quality of the writing English is quite low, with several parts of the manuscript that are almost impossible to understand. This must be addressed by almost rewriting the whole paper and a revision by a native speaker is highly recommended.
2. The novelty of the study should be addressed, for example, in the final part of the Introduction section.
3. In section 2.1.1., the authors must report the percentages of cell adhesion.
4. Figure 3C. What is the difference between the images (HE and Alcian Blue) on the left and right? In the group of images on the right, both were described as “HE”.
5. Figure 3 and section 2.1.3. To assess the chondrogenic differentiation, other assays such the quantification of the amount of GAG and immunohistochemical analysis for collagen II and aggrecan should be performed.
6. Figure 4. The image needs to be restructured to allow the reading in graphs B and D.
Peer-review bioengineering – 2416243
The research manuscript entitled “Peptide E7 enables the BMSCs Adhesion and Promote the Chondrogenic Differentiation of BMSCs Via LncRNA H19/miR675 axis” seems an interesting study on the role of peptide E7 in the adhesion, proliferation and chondrogenic differentiation. The manuscript topic fits well within the scope of the journal Bioengineering (ISSN 2306-5354). However, in this current state, the manuscript cannot be considered for publication.
Issues:
1. The quality of the writing English is quite low, with several parts of the manuscript that are almost impossible to understand. This must be addressed by almost rewriting the whole paper and a revision by a native speaker is highly recommended.
2. The novelty of the study should be addressed, for example, in the final part of the Introduction section.
3. In section 2.1.1., the authors must report the percentages of cell adhesion.
4. Figure 3C. What is the difference between the images (HE and Alcian Blue) on the left and right? In the group of images on the right, both were described as “HE”.
5. Figure 3 and section 2.1.3. To assess the chondrogenic differentiation, other assays such the quantification of the amount of GAG and immunohistochemical analysis for collagen II and aggrecan should be performed.
6. Figure 4. The image needs to be restructured to allow the reading in graphs B and D.
Author Response
- The quality of the writing English is quite low, with several parts of the manuscript that are almost impossible to understand. This must be addressed by almost rewriting the whole paper and a revision by a native speaker is highly recommended.
Response: Thanks for your comments. According to your suggestion, the revised manuscript has undergone English language editing by MDPI. The text has been checked for correct use of grammar and common technical terms, and the quality of the writing English is improved. The English editing certificate was also submitted.
- The novelty of the study should be addressed, for example, in the final part of the Introduction section.
Response: Thanks for your comment. The novelty of the study had been addressed in the final part of the Introduction section: “Through the measurements of the amount, viability and phenotype of BMSCs that adhered to free KGM, E7-KGM and RGD-KGM, we have found the positive role of E7 peptide in the function of BMSC adhesion and chondrogenic differentiation of BMSCs when compared to RGD peptide. These findings could be utilized to enhance the effectiveness of endogenous BMSC homing and promote the chondrogenic differentiation effect of BMSCs in cartilage tissue engineering. This may offer valuable clues for developing novel strategies for cartilage regeneration and repair.” (Line 68-74)
- In section 2.1.1., the authors must report the percentages of cell adhesion.
Response: Thanks for your comment. We have added the percentages of cell adhesion in the section 2.1: “The percentages of cell adhesion were 29.58±7.54% for E7-KGM, 36.26±13.17% for RGD-KGM, 10.00±3.31% for M7-KGM, and 7.92±2.53 for Blank KGM.” (Line 85-87). The related method was added in the section 4.3 in the revised manuscript :“For calculating the percentages of cell adhesion, we shake the culture dish gently to make the unadhered BMSCs equably suspended in the medium after co-culture of BMSCs and KGMs in serum-free culture medium for 12 hours. Whereafter, we transfer all of the medium with unadhered BMSCs suspending inside to do the cell counting in the cell counting plate, the amount of unadhered BMSCs in the medium was calculated. The number of adherent cells in the KGM was calculated by the total number of implanted cells minus the number of unadhered BMSCs. The percentages of cell adhesion are defined as the total number of cells divided by the number of adherent cells.” (Line 374-382).
- Figure 3C. What is the difference between the images (HE and Alcian Blue) on the left and right? In the group of images on the right, both were described as “HE”.
Response: Thanks for your comments. We feel sorry for not making the difference between the images on the left and right. The images on the left indicates the HE staining and Alcian blue staining at day 5, and the images on the right indicated the staining at day 14, we have corrected the figures in the revised manuscript.
- Figure 3 and section 2.1.3. To assess the chondrogenic differentiation, other assays such the quantification of the amount of GAG and immunohistochemical analysis for collagen II and aggrecan should be performed.
Response: Thanks for your comments. In this study, the expression levels of Col II, Acan and Sox 9, and Alcian blue staining for glycosaminoglycans were performed to assess the chondrogenic differentiation, which were the most common assays for evaluating the effectiveness of chondrogenic differentiation[1-3]. These assays can already help us compare and analyze the chondrogenic differentiation between different groups. Of course, we totally agree with your opinion that other assays such the quantification of the amount of GAG and immunohistochemical analysis for collagen II and aggrecan can more comprehensively reflect the chondrogenic differentiation, and we will add these assays in our future cartilage repair study.
- Figure 4. The image needs to be restructured to allow the reading in graphs B and D.
Response: Thanks for your comments. According to your suggestion, we have restructured the Figure 4 to allow the reading in graphs B and D in the revised mauscript.

Reviewer 2 Report
Shi et al. reported the investigation of the BMSCs adhesion on the glucomannan microspheres modified with an E7 heptapeptide. The authors evaluated the effect of E7 peptide on BMSCs viability, promotion of the chondrogenic differentiation of BMSCs, mRNA expression and inhibition of H19/mi-675. Despite the good results and appropriate discussion, the structure of the manuscript is poor and the manuscript should be carefully revised. Specific comments are below:
1. Please, put the figures and table at places where they are mentioned instead of collecting them in a separate section.
2. Section 2.1.1. starts with the description of the modification of microparticles with E7 peptide. The description in lines 78-80 is unclear. Looking on this it seems that authors conjugate peptide through the thiol to epoxide on the surface of KGMs. However, looking in figure 7A it is not so. Please, revise the description to make it in agreement with the scheme.
3. Figure 7A is not mentioned in the text. I recommend to move it to section 2.1.1 and make it a Figure 1. It is logic to start this section with the description of KGMs preparation and modification with E7 and support this with a reaction scheme.
4. Line 82. What is “electronic modified”?
5. Figures 1-3. Scale bars are missed in some images. Please provide them.
6. Figures 1-3. Please provide in figure legend the type of microscopy used for the analysis in each case.
7. Figure 1. It would be better to provide the explanations to the used abbreviations in the figure legend.
8. Table 1 was not mentioned in the text. In my opinion it should be moved to the Experimental part.
9. Section 4. A separate section summarizing the materials, chemicals, biologicals, supplements and their suppliers is necessary.
Author Response
- Please, put the figures and table at places where they are mentioned instead of collecting them in a separate section.
Response: Thanks for your comment. According to your suggestion, we have adjusted the position of all figures and table to make them at places where they are mentioned in the revised manuscript.
- Section 2.1.1. starts with the description of the modification of microparticles with E7 peptide. The description in lines 78-80 is unclear. Looking on this it seems that authors conjugate peptide through the thiol to epoxide on the surface of KGMs. However, looking in figure 7A it is not so. Please, revise the description to make it in agreement with the scheme.
Response: Thanks for your comments. We feel sorry for not describing the modification of microparticles with E7 peptide clearly. In fact, we conjugated peptide through the thiol to epoxide on the surface of KGMs and the figure 7A have make some misunderstandings. We have revised the scheme in figure 7A (showed as Figure 1A in the revised manuscript).
- Figure 7A is not mentioned in the text. I recommend to move it to section 2.1.1 and make it a Figure 1. It is logic to start this section with the description of KGMs preparation and modification with E7 and support this with a reaction scheme.
Response: Thanks for your comments. According to your suggestion, we have moved the Figure 7A to section2. and showed as Figure 1A. It was mentioned in the revised manuscript as: “KGMs with an average diameter of 200 µm were prepared according to a previous study. When E7 peptide was added into the system, the sulfydryl group of E7 peptide combined with the epoxy group on the surface of KGMs, leading to the formation of E7-coated KGMs (E7-KGM) (Figure 1A).” (Line 79-82)
- Line 82. What is “electronic modified”?
Response: Thanks for your comments. In the previous manuscript, “electronic modified” indicates the amino-modified KGM microcarriers by an amination process with DEAE·HCl. To describe more appropriately, we have revised “electronic modified” to “electric charge modified”.
- Figures 1-3. Scale bars are missed in some images. Please provide them.
Response: Thanks for your comments. We feel sorry for not marking scale bars in some images. According to your suggestion, we have checked all images and have added scale bars in Figure 1B, 2A, 2B and Figure 3C.
- Figures 1-3. Please provide in figure legend the type of microscopy used for the analysis in each case.
Response: Thanks for your comments. According to your suggestion, we have added the type of microscopy used for the analysis in each case in figure legend in the revised manuscript.
- Figure 1. It would be better to provide the explanations to the used abbreviations in the figure legend.
Response: Thanks for your comments. According to your suggestion, we have provided the explanations to the used abbreviations in the figure 1 legend. It was mentioned in the revised manuscript as: “Fig. 1. (A) Scheme illustrating the preparation process of E7-KGMs. (B) BMSCs were co-cultured with EC-KGMs, E7-KGMs, RGD-KGMs, M7-KGMs and free KGMs for 1, 3 and 5 days, and the adhesion of BMSCs on KGMs was observed under optical inverted microscope (n=4). White scale bar =1 mm, and black scale bar =200 µm. EC-KGMs, electronic charge modified konjac gluco-mannan microsphere ; E7-KGMs, E7 (EPLQLKM) modified konjac glucomannan microsphere; RGD-KGMs, RGD peptide (Arg-Gly-Asp) modified konjac glucomannan microsphere; M7-KGMs, M7(MLKPLEQ) modified konjac glucomannan microsphere.” (Line 99-105)
- Table 1 was not mentioned in the text. In my opinion it should be moved to the Experimental part.
Response: Thanks for your comments. According to your suggestion, we have moved the Table 1 to the Experimental part in the revised manuscript. (Line 424)
- Section 4. A separate section summarizing the materials, chemicals, biologicals, supplements and their suppliers is necessary.
Response: Thanks for your comments. In the manuscript, the name and suppliers of materials, chemicals, biologicals, and supplements were described at where they are mentioned, such as, LIVE/DEAD staining (Invitrogen, Carlsbad, CA, USA), Medium (Cyagen Biosciences), and SEM (S-4800 field-emission scanning electron microscope; Hitachi, Tokyo, Japan) as the publications in this Journal.[3,4]

Reviewer 3 Report
The article by Shi et al. reported the role of E7 in the adhesion and viability of BMSCs as well as the pro-chondrogenic effect of E7 on BMSCs through H19/miR765 axis, suggesting that this system for cartilage regeneration. However, some issues should be clarified:
- introduction: the chondrogenesis should be reported, describing the main gene involved, with particular regard to that reported in the experiments; line 88 bioacive should be bioactive;
- for all histograms the statistical significancy should be revised, keeping particular attention to evaluate all treatments; i.e. fig 2D reported the p only for the comparison between E7 and EC, but surely RGD compared to EC is statistically significant. Thus, then results and discussion should be reformulated. The authors should detail the statistical test used for each figure/experiment.
- a better resolution for acan western blotting will be suitable.
- it is important to remember in the text that cartilage for regeneration requires external interventions: this will further sustain the importance of the paper
Language is okay
Author Response
- introduction: the chondrogenesis should be reported, describing the main gene involved, with particular regard to that reported in the experiments; line 88 bioacive should be bioactive;
Response: Thanks for your comments. We have added the main gene in the introduction: “Chondrogenesis of MSCs featured by expression of genes including col II, acan, sox9 and secretion of extracellular matrix could contribute to the cartilage regeneration.” (Line 43-44). We have changed bioactive to bioactive.
- for all histograms the statistical significancy should be revised, keeping particular attention to evaluate all treatments; i.e. fig 2D reported the p only for the comparison between E7 and EC, but surely RGD compared to EC is statistically significant. Thus, then results and discussion should be reformulated. The authors should detail the statistical test used for each figure/experiment.
Response: Thanks for your comments. We feel sorry for neglecting to annotate statistical significancy in some histograms, and we have detailed all the statistical test used for each figure. In the revised manuscript, we have added the statistically in Figure 2C,3A,3B and reformulated the results and discussions.
- a better resolution for acan western blotting will be suitable.
Response: Thanks for your comment. We have replaced a better resolution edition of acan western blotting in the revised manuscript.
- it is important to remember in the text that cartilage for regeneration requires external interventions: this will further sustain the importance of the paper
Response: Thanks for your comment. We totally agree your opinion that cartilage for regeneration requires external interventions, and we have added the importance of this paper in the Introduction section: “These findings could be utilized to enhance the effectiveness of endogenous BMSC homing and promote the chondrogenic differentiation effect of BMSCs in cartilage tissue engineering. This may offer valuable clues for developing novel strategies for cartilage regeneration and repair.” (Line 71-74)

Round 2
Reviewer 1 Report
The manuscript was considerably improved after this round of revisions, with almost all the comments/suggestions addressed properly.
Figure 4 can still be improved to allow a better reading and there is still room to improve the quality of the written English.
The manuscript was considerably improved after this round of revisions, with almost all the comments/suggestions addressed properly.
Figure 4 can still be improved to allow a better reading and there is still room to improve the quality of the written English.
Author Response
Response: Thanks for your comments. According to your suggestion, we have improved the quality of Figure 4 and the written English for a better reading in the revised manuscript.
Reviewer 3 Report
The authors addressed all my concerns.
Author Response
Thank you for your recognition.